# COVID-19 Infection in Chronic Kidney Disease Patients in Bulgaria: Risk Factors for Death and Acute Kidney Injury

**DOI:** 10.3390/jpm12101676

**Published:** 2022-10-09

**Authors:** Rumen Filev, Lionel Rostaing, Mila Lyubomirova, Boris Bogov, Krassimir Kalinov, Dobrin Svinarov

**Affiliations:** 1Department of Nephrology, Internal Disease Clinic, University Hospital “Saint Anna”, 1750 Sofia, Bulgaria; 2Medical Faculty, Medical University Sofia, 1431 Sofia, Bulgaria; 3Nephrology, Hemodialysis, Apheresis and Kidney Transplantation Department, Grenoble University Hospital, 38700 Grenoble, France; 4Grenoble Alpes University, 38400 Grenoble, France; 5Head Biometrics Group, Comac-Medical Ltd., 1404 Sofia, Bulgaria; 6Department of Clinical Laboratory, University Hospital “Alexandrovska”, 1431 Sofia, Bulgaria

**Keywords:** COVID-19, chronic kidney disease, acute kidney injury, risk factors, death

## Abstract

Regarding COVID-19 infection, Bulgaria has one of the lowest rates of vaccination in Europe, and its COVID-19-related mortality rate has been one of the highest in the European Union. Chronic kidney disease (CKD)-COVID-19 patients are at higher risk of developing acute kidney injury (AKI) and death after hospital admission. This single-center prospective cohort study from Bulgaria included 120 in-patient COVID-19 subjects of whom 70 had CKD and 50 normal renal function. Diabetes mellitus, hypertension, obesity, and cardiovascular disease were statistically more prevalent in the CKD group as compared to the non-CKD group. At admission, D-dimer, creatinine, and urea levels were significantly higher in the CKD group, whereas estimated glomerular-filtration rate was significantly lower as compared to the non-CKD patients. During hospitalization, 23 patients (19.1%) died, of which 19 were in the CKD group (*p*-value = 0.0096); in addition, 38 developed AKI (31.6%), of which 31 were in the CKD group (*p*-value = 0.0006). Using binary logistic regression, being male, having experienced AKI, and not having been treated with remdesivir were independent risk factors for COVID-19-induced mortality. Regarding risk of AKI, having had COVID-19-related symptoms for more than 6 days before admission, having CKD at baseline, and having not received remdesivir therapy were independent predictive factors for developing AKI after admission.

## 1. Introduction

The SARS-CoV-2-related coronavirus 2019 (COVID-19) pandemic has killed millions of people worldwide (https://en.wikipedia.org/wiki/Template:COVID-19_pandemic_data (accessed on 25 August 2022)). From a medical perspective, the crisis in Bulgaria has not differed from the rest of the world; however, the statistics have revealed disturbing tendencies; i.e., the percentage vaccinated in Bulgaria has been one of the lowest in Europe, and its mortality rate has been one of the highest in the European Union [1].

Although we know that the main feature of SARS-CoV-2 is diffuse alveolar damage, its involvement in other organs has yet to be fully determined [2]. It was shown in 2005 that, amongst patients presenting with coronavirus-associated severe acute respiratory syndrome (SARS), acute kidney injury (AKI) developed in 6.7% at a median duration of 20 days (range 5–48 days) after the onset of viral infection despite a normal plasma creatinine level at first clinical presentation [3]. In addition, the mortality rate was significantly higher among patients with SARS and AKI compared with those with SARS and no renal impairment (91.7% vs. 8.8%) (*p* < 0.0001) [3].

AKI caused by COVID-19 is a common complication in hospitalized COVID-19-positive patients and has been reported in several countries and regions, resulting in high patient mortality [4]. It is also associated with increased severity of illness, prolonged duration of hospitalization, and a poor prognosis [5]. AKI usually occurs early in the disease and in parallel with lung damage [6]. In addition, the most common comorbidities in patients with COVID-19 suffering from AKI are a history of diabetes, hypertension, and/or hyperlipidemia [5]. COVID-19-induced kidney injury is affected by several factors, including direct kidney injury mediated by the combination of virus and angiotensin-converting enzyme 2, immune-response dysregulation, a cytokine storm driven by SARS-CoV-2 infection, organ interactions, hypercoagulable state, and endothelial dysfunction [7]. 

The prevalence of chronic kidney disease worldwide is high [8]. A recent study has shown that COVID-19-CKD patients presenting with AKI had a lower survival rate than those with AKI or CKD only [9]. In addition, multiple logistic regression analyses indicated that the predictors of AKI in COVID-19 patients included complications, such as respiratory failure and acute myocardial injury, and higher creatinine and procalcitonin levels during hospitalization [9]. In addition, children with CKD presenting with moderate-to-severe COVID-19 or with nephrotic syndrome relapse are at risk of severe complications, including severe AKI and mortality [10].

Because of the recent findings on the detrimental effect of CKD and AKI on COVID-19-hospitalized patients, we conducted a prospective study that included 120 COVID-19 patients in a university hospital in Bulgaria. The aim was to explore the risk factors for COVID-19-related mortality in CKD patients as primary endpoint and as secondary to find out the risk factors for AKI. 

## 2. Patients and Methods

This single-center study was performed at Alexandrovka’s Hospital in Sofia (Bulgaria) and was conducted between 1 February 2021 and 31 March 2021. We included consecutive patients that were confirmed to have a COVID-19 infection and who were admitted into hospital after a positive PCR-test for SARS-CoV-2. All cases were confirmed using the reverse transcription-polymerase chain reaction from the combined throat/nose samples [2,11]. Only patients aged >18 years were included in this study. Patients with a urinary-tract infection were also excluded. All the patients were Caucasians. 

At the starting date of the study, there were in total 13,656 Bulgarian people vaccinated with two doses against the SARS-CoV-2 virus, and in the end of March 2021, the total number was 95,003 Bulgarian people who had two doses. By the data according to the National Statistical Institute of Bulgaria, this represents 0.20% of the population of Bulgaria at the beginning of the study [1]. From all 120 patients that were included in the study, none of them was vaccinated against SARS-CoV-2. 

There were 120 patients: 70 had a history of chronic kidney disease (by definition in KDIGO), and none of them had end-stage renal disease. Some patients had eGFR greater than 60 mL/min/1.73 m^2^, including five kidney transplant recipients who had functional renal allograft for more than 5 years. Indeed, CKD was based on previous eGFR that was calculated on results well before admission in order to exclude false results based on the level of creatine on the day of the admission. The other 50 patients made up the control group: they had no history of kidney disease and had normal levels of serum creatinine (normal level of creatinine for females 44–80 µmol/L and for males 62–106 µmol/L; GFR CKD-EPI 2021 over 60 mL/min/1.73 m^2^). 

Data on patients’ gender, age, comorbidities, and laboratory results from blood and urine samples, taken at admission into the emergency room, were collected. Laboratory values on serum creatinine, calculated eGFR (by the CKD-EPI 2021 formula) [12], highly sensitive inflammatory markers, D-dimer, fibrinogen, total protein, albumin, blood-cell counts, and changes in urine parameters (hematuria, proteinuria) were collected. At 7–10 days after admission, further laboratory analyses were conducted to assess the follow-up period.

Comorbidities included hypertension, obesity (defined by a body mass index of ≥30 kg/m^2^), diabetes mellitus, vascular disease, and chronic kidney disease (CKD). CKD had to be already diagnosed with a medical history. Kidney function was estimated by CKD-EPI 2021. c [13].

Cases of pneumonia were defined using the CT-scan Severity Score Index, a scoring system that assesses lung changes and involvement with COVID-19 based on approximate estimation of the pulmonary areas involved [14].

In-hospital treatment was implemented following the official protocol published by the Bulgarian Ministry of Health [15]. Information on medications that the patients were taking before hospitalization were not available for all patients: thus, only medications received during hospital treatment were entered into the database. On the first day of admission, 100% of patients received antibiotics (levofloxacin, azithromycin, ceftriaxone, or meropenem) with dosages depending on the condition of the patient. One hundred percent of patients received low-molecular-weight or intravenous heparin or glucocorticoids (dexamethasone or methylprednisolone), whereas antiviral treatment, i.e., remdesivir, was given to 45% of patients. Remdesivir was prescribed to patients who showed severely impaired general condition (ICU patients, patients with high lung severity score from the CT-scan, oxygen saturation (SpO2) ≤ 92% on room air, or requiring supplemental oxygen) or patients who were at high risk of progressing to severe COVID-19 because of their comorbidities. Moreover, another factor at that short period of time was that there were not enough amounts of remdesivir delivered to the hospital. 

These were the most common medications given to patients with a COVID-19 infection. Throughout the in-hospital treatment, the percentage of patients that were treated with diuretics and antifungal drugs became higher after the first week.

The study is prospective and was conducted according to the guidelines of the Declaration of Helsinki and approved by the ethical committee KENIMUS of the Medical University of Sofia, Bulgaria, with Protocol №12/31.05.2022. All data are available upon request to the corresponding author. The primary endpoint was the risk factors for COVID-19-related mortality in CKD patients, and the secondary endpoint was to assess the risk factors for AKI in patients with COVID-19 infection.

## 3. Statistical Analyses

-Categorical parameters were analyzed for absolute and relative (percentage) frequencies. Continuous parameters were analyzed for arithmetic mean and standard deviation (SD) and median, minimum, and maximum values. The distribution of continuous parameters was checked for normality using the Shapiro–Wilk test.-Methods for testing the post hoc hypotheses:-To compare continuous parameters in two related (paired) groups, the Wilcoxon two-sample test was applied, and approximation of the Students’ *t*-statistic (*t*-approximation) with a continuity correction of 0.5 used to determine statistical significance. In addition, the sign test was used where appropriate.-Comparisons between independent (unrelated) groups (CKD vs. no CKD) were made after adjustment for baseline differences. Continuous variables were analyzed using a general linear model with a baseline value as the covariate and the group as a factor. Cohran–Mantel–Haenzel statistics were used to compare groups for categorical variables.

### Method Used for Data Modelling

Binary logistic regression was used to model the relation between output (death and AKI) as a dependent variable and the main parameters. Two separate models are presented: one for mortality and one for AKI. In addition, odds ratios and forest plots are shown. For decision making, a significance level of 5% was used. 

The SAS^®^ package ver. 9.4 (SAS Institute Inc., SAS 9.4 Help and Documentation, Cary, NC: SAS Institute Inc., 2015–2022) was used for the calculations and the graphical presentations.

## 4. Results

Of the 120 confirmed COVID-19 patients, 70 (58.3%) had a history of CKD. In the control group, there were 50 patients (41.7%). The overall median age of the patients was 65.7 years, and the gender ratio was 50% male and female in both groups. The median time for the duration of the symptoms before hospitalization was 6 days, with ranges from a minimum of 1 day and a maximum of 20 days before hospitalization. Of the 120 patients, ~35% were febrile—temperature over 38 °C. The total percentage of patients reported to have a comorbidity (arterial hypertension, diabetes, obesity, cardiovascular disease) with CKD or without CKD was 73.3%. Patients who had at least one or more comorbidities were divided as follows: obesity—14.2%, hypertension—65.8%, diabetes type II—34.2% (no patients had type I diabetes), and cardiovascular disease—35.8%. A comparison between the demographics and baseline parameters was performed (Table 1). 

The data show that diabetes mellitus, arterial hypertension, obesity, and cardiovascular disease were statistically more prevalent in the CKD group as compared to the non-CKD group.

Interestingly, on admission, 23 patients (19.1%) had normal levels of CRP, 66 patients (55%) had normal range of serum ferritin, 36 patients (30%) had normal levels of D-dimers, and finally, for fibrinogen, 60 patients (50%) had normal levels. 

A first comparison was performed between all the baseline laboratory values across the two groups (Table 2). There were a few significant differences between the two groups (i.e., non-CKD patients and CKD patients) at admission: i.e., serum creatinine level, eGFR, urea, D-dimer, hemoglobin, proteinuria, and hematuria.

Regarding D-dimer levels on admission, overall, they were elevated in both groups (72% patients in total), whilst at the follow-up 7–10 days after admission, D-dimer levels were still increased in 46% of the patients. 

Overall, creatinine level on admission was elevated in 58.3% of cases; i.e., normal ranges for males are 62–106 μmol/L and for females 44–80 μmol/L, and eGFR was <60 mL/min/1.73 m^2^ in 50% of patients. The mean value of eGFR on admission was 82.3 mL/min/1.73 m^2^ for the non-CKD group and 49.5 mL/min/1.73 m^2^ for the CKD group (*p*-value ≤ 0.001). These values were higher in both groups in the 7–10-day follow-up results: i.e., 92.2 mL/min/1.73 m^2^ for the non-CKD and 59.0 mL/min/1.73 m^2^ for the CKD group (*p*-value = 0.001). In addition, patients that had AN elevated creatinine level (for males over 106 μmol/L and for females over 80 μmol/L) in the baseline data totaled 58.3%, whereas in the 7–10-day follow-up after hospitalization, the percentage was lower (40.8%). In total, three patients needed renal replacement therapy: two patients from the CKD group and one from the non-CKD group. For all three patients, the hemodialysis was performed in the ICU.

Urine samples at admission showed 39 patients had proteinuria, and of them, 87.1% had 1+ proteinuria, and the others had >1+. At 7–10 days after admission, the number of cases of proteinuria was higher than at admission: i.e., 83 patients, of which 69.8% had 1+ proteinuria, and the others had over 1+. We also assessed hematuria: at admission, there were 22 cases, of which 54.5% had only 1+ hematuria, whereas at days 7–10 after admission, there were 37 cases, of which only 27% had hematuria higher than 1+ (*p*-value 0.05). Fewer patients had hematuria compared to proteinuria. The incidence of ureteral catheterization was quite low (<8.3%), which was expected to have low impact on the urine analysis.

Within-group comparisons showed there were relatively significant changes between those at admission and at follow-up. In CKD patients, there was statistically significant decrease in serum creatinine, D-dimer, hemoglobin, proteins, CRP, and ferritin, whereas there was statistically significant increase in eGFR, leukocytes, platelets, and oxygen saturation. In non-CKD patients, we observed the same findings except for D-dimer, albumin, and ferritin, whose changes were not statistically significant (Table 3).

Acute kidney injury was a frequent event; i.e., it occurred in 38 patients (31.6%), of which 31 were in the CKD group (44.3% of CKD patients) (*p*-value = 0.0006). All patients that had AKI could be divided into three stages using the KDIGO criteria. Patients within the control group had only seven cases of AKI. In the CKD group, 30 patients (i.e., 42.8%) had AKI Stage 3. In the CKD group, those that had CKD Stage 2 had the highest incidence of AKI—36.8% from the 38 patients. Overall, within our cohort of 120 patients, in-hospital mortality was 19.1% (23 patients), including the five transplanted recipients, and of these, 66.6% of the patients had AKI (19 patients; *p*-value = 0.0096). Amongst overall patients that did not survive COVID-19, 100% had kidney disease (CKD and/or AKI)—17.2% of the non-CKD group had AKI, 30.4% patients had CKD (any stage by KDIGO), and 52.1% of the patients had AKI and CKD (any stage by KDIGO). 

All the patients who did not survive the SARS-CoV-2 infection were patients in the ICU. All of them had bilateral pneumonia and were mechanically ventilated. The cause of death was the advanced pneumonia, and none of the patients had records for any kind of vascular incident. 

We then analyzed the risk factors that may have led to death or AKI. Table 4 shows logistic regression results for the mortality risk factors. Statistically significant factors for mortality in our cohort were male gender, AKI, and not being treated with remdesivir. These results were confirmed by assessing the odds ratio: the results can be seen in Table 5 and Figure 1. AKI was recognized as a negative prognostic factor for COVID-19 infection; i.e., it increased death (OR = 5.26 (95% CI: 1.15–24.05); *p* = 0.03).

The same analysis was performed for the risk factors for AKI. Table 6 shows the logistic regression for the risk factors for AKI. The statistically significant factors for the incidence of AKI were having an eGFR equal to or less than 45 mL/min/1.73 m^2^, with symptoms for 6 or more days before hospitalization, and not having received remdesivir. The same results were confirmed using the odds ratio (Table 7 and Figure 2).

## 5. Discussion

In our study, we found that being male, having experienced AKI, and not having been treated with remdesivir were independent risk factors for COVID-19-induced mortality. Regarding the risk of AKI, we found that having had COVID-19-related symptoms for more than 6 days before hospital admission, having CKD at baseline, and having not received remdesivir therapy were independent predictive factors for developing AKI after admission for COVID-19 infection.

We found that remdesivir was an independent protective factor against COVID-19-related mortality and for COVID-19-related AKI. Remdesivir is an antiviral drug [16]. The PINETREE study has shown that, among non-hospitalized COVID-19 patients that were at high risk for COVID-19 progression, a 3-day course of remdesivir had an acceptable safety profile and resulted in an 87% lower risk of hospitalization or death than a placebo [17]. However, the DisCoVeRy study reported no clinical benefit from the use of remdesivir given to patients admitted into hospital for COVID-19, that were symptomatic for more than 7 days, and required oxygen support [18]. A recent meta-analysis stated that “remdesivir could still be effective in early clinical improvement; reduction of early mortality and avoiding high-flow supplemental oxygen and invasive mechanical ventilation among hospitalized COVID-19 patients” [19].

We observed that male gender was an independent risk factor for COVID-19-related mortality. It is well-known that the COVID-19-related mortality is associated with gender; i.e., male COVID-19 patients have a significantly higher mortality rate than female COVID-19 patients [20,21]. Women appear to be less prone to severe forms of the disease, and their mortality is lower than for men. The role of female hormones in the modulation of inflammation may be the reason behind this gender difference [22].

Finally, we found that COVID-19-related mortality was impacted by AKI; i.e., AKI was an independent risk factor for mortality. Indeed, Ng et al. showed, in a cohort of 9657 patients admitted with COVID-19, that the AKI incidence rate was 38.4/1000 patient-days [23]. Incidence rates of in-hospital death among patients without AKI, with AKI not requiring dialysis (AKI stages 1–3), and with those with AKI receiving dialysis (AKI 3D) were 10.8, 31.1, and 37.5/1000 patient-days, respectively. After adjusting for demographics, comorbid conditions, and illness severity, the risk for death remained higher among those with AKI Stages 1–3 (adjusted HR, 3.4 (95% CI, 3.0–3.9)) and AKI 3D (adjusted HR, 6.4 (95% CI, 5.5–7.60) compared to those without AKI [23]. Alfano et al. recently reported that COVID-19-associated AKI was independently associated with in-hospital death (hazard ratio (HR) = 4.82; CI 95%, 1.36–17.08) compared to non-AKI patients [24].

AKI is very prevalent in COVID-19-hospitalized patients. Hence, in a UK cohort of 1248 inpatients, 487 (39%) experienced AKI (51% Stage 1, 13% Stage 2, and 36% Stage 3) [25]. Regarding AKI risk factors, we found that remdesivir was significantly associated in COVID-19 patients with a decreased risk of AKI (OR 3.73, CI 1.26-11.01; *p* = 0.01). Indeed, it has been shown that patients with COVID-19 and receiving steroids and remdesivir had a reduced risk of AKI in propensity matched analysis (17.5% vs. 23.4%, respectively, *p* = 0.001) [26]. In our study, we acknowledge that remdesivir therapy was administered in an unusual way; however, this was a real-life study and because at some point, there was a shortage of remdesivir in Bulgaria, it was decided to give it to only the most seriously affected patients. We recognize that this might have biased the results.

We observed that having had COVID-19-related symptoms before admission for more than 6 days was a significant risk factor for developing AKI (OR: 2.97, CI 0.98–8.99, *p* = 0.053). Indeed, Diebold et al. showed that AKI developed after a median of 9 days (interquartile range (IQR) 5–12) after the first symptoms and a median of 1 day (IQR 0–5) after hospital admission [27]. Recently, Gur et al. observed that despite comparable disease severity at presentation, COVID-19 patients with CKD had significantly more AKI events and required more renal replacement therapy during hospitalization than control COVID-19 patients did. Thus, the odds ratio for AKI was 5.8 (95%CI 3.8–8.7; *p* < 0.001) in CKD group vs. control group [28]. In a recent meta-analysis, Cai et al. found that chronic kidney disease was an independent risk factor for developing AKI in hospitalized COVID-19 patients, i.e., OR = 4.56 [29]. Krishnasamy et al. found that CKD children presenting with moderate-to-severe COVID-19 or in nephrotic syndrome relapse are at risk of severe AKI [10].

Our study has major implications for COVID-19 patients in Bulgaria. Firstly, the shorter the time with COVID-19-related symptoms before admission, the lower the risk of developing AKI. Therefore, general practitioners and would-be patients should be aware of that. Secondly, we found that remdesivir therapy is able to significantly decrease COVID-19-related death and AKI after hospital admission. This might be explained by the fact that in a country where the anti-COVID-19 vaccination rate is the lowest in the EU, would-be patients have no immunity, and early implementation of antiviral therapy is efficient. Thirdly, as we found that as AKI was significantly associated with COVID-19-related death, it is of utmost importance to prevent AKI, especially in those with underlying CKD. This implies stopping at hospital admission every potential nephrotoxic therapy but also avoiding the use of potential nephrotoxic agents during the hospital stay. If such kind of nephrotoxic drugs must be used, the dosing should be adapted to renal function, and also, blood monitoring must be implemented. 

## 6. Conclusions

In conclusion, we did not find that CKD per se was a risk factor COVID-19-related death or AKI after patient hospitalization. Conversely, developing AKI was a significant risk factor of in-hospitalization death. Therefore, preventing AKI is of the utmost importance. Finally, in a population where vaccine protection against COVID-19 is very low, the use of remdesivir was able to significantly decrease COVID-19-related death and AKI. 

## Figures and Tables

**Figure 1 jpm-12-01676-f001:**
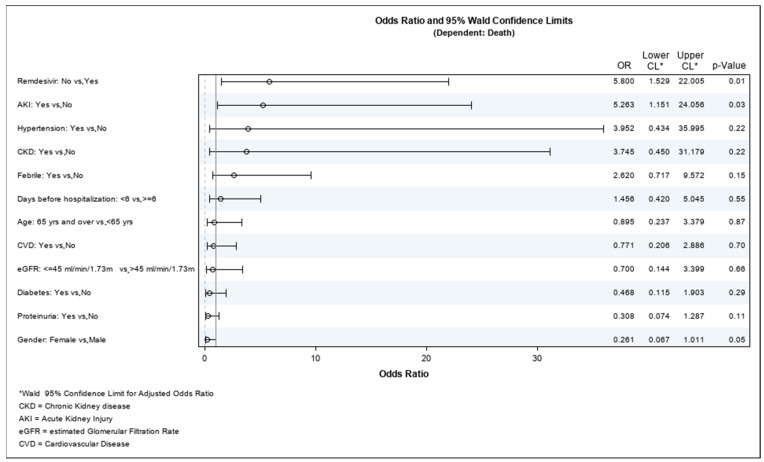
Logistic regression for mortality.

**Figure 2 jpm-12-01676-f002:**
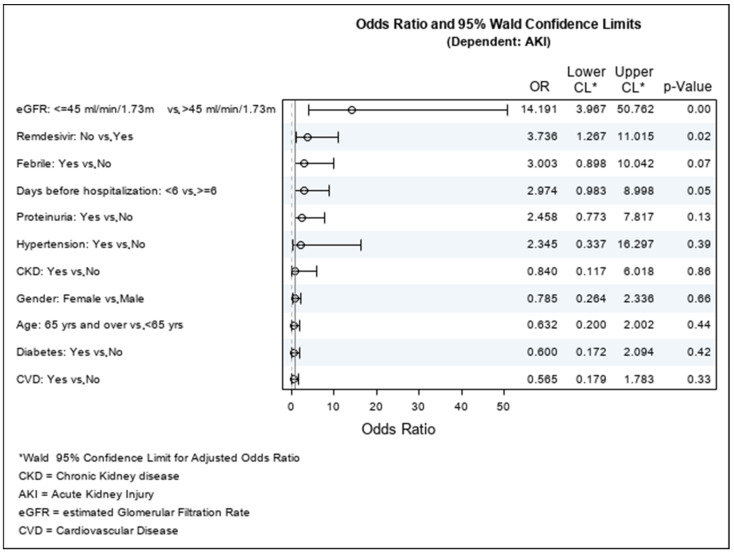
Odds ratio for acute kidney injury.

**Table 1 jpm-12-01676-t001:** Baseline demographic of the study population.

		CKD	Non-CKD	*p*-Value
**Number of patients**		70	50	
Age of the patients (years)	Mean (SD)	56.8 (16.00)	65.9 (12.56)	0.0007
Gender (n%)	Female	21 (42.0%)	39 (55,7%)	0.13
	Male	29 (58.0%)	31 (44,3%)	
Race (n%)	Caucasian	50 (100%)	70 (100%)	
Obesity (n%)	No	49 (98.0%)	54 (77,1%)	0.001
	Yes	1 (2.0%)	16 (22.9%)	
Cardiovascular disease (n%)	No	41 (82.0%)	36 (51.4%)	0.0006
	Yes	9 (18.0%)	34 48.6%)	
Diabetes (n%)	No	49 (98.0%)	30 (42.9%)	<0.0001
	Yes	1 (2.0%)	40 (57.1%)	
Hypertension (n%)	No	38 (76.0%)	3 (4.3%)	<0.0001
	Yes	12 (24.0%)	67 (95.7%)	
Use of ACEI/ARB n (%)	No	40 (80.0%)	43 (61.4%)	0.08
	Yes	10 (20.0%)	26 (37.1%)	

Abbreviations: CKD, chronic kidney disease; ACEI/ARB, angiotensin-converting enzyme inhibitor; ARB, angiotensin receptor blocker.

**Table 2 jpm-12-01676-t002:** Baseline comparison between the two groups according to biological parameters.

Parameter	Statistical Analysis	CKD Patients	Non-CKD Patients	*p*-Value
Number of patients	N	70	50	
Creatinine (μmol/L)	Median (Ranges)	119 (57.0–930.0)	79 * (50.0–1295.0)	<0.001
eGFR (mL/min)	Mean (SD)	49.5 (24.07)	82.3 (27.77)	<0.001
Urea (mmol/L)	Median (Ranges)	9.2 (2.7–75.2)	5.5 * (3.0–82.3)	<0.001
D-Dimer (mg/L FEU)	Median (Ranges)	0.9 (0.3–10.7)	0.5 (0.3–8.1)	<0.001
Fibrinogen (g/L)	Mean (SD)	4.6 (1.58)	4.3 (1.52)	0.52
Hemoglobin (g/L)	Mean (SD)	133.7 (19.61)	142.5 (15.99)	0.01
Leucocytes (×10^3^/mm^3^)	Median (Ranges)	7.4 (2.5–21.2)	6.8 (2.9–35.3)	0.31
Lymphocytes (×10^3^/mm^3^)	Mean (SD)	1.3 (0.61)	1.4 (0.68)	0.71
Platelet count (×10^3^/mm^3^)	Mean (SD)	247.6 (107.31)	246.0 (85.14)	0.92
Protein (g/L)	Mean (SD)	62.7 (7.09)	64.1 (6.23)	0.09
CRP (mg/L)	Median (Ranges)	50.8 (0.5–320.6)	29.8 (0.1–217.6)	0.06
Ferritin (μg/L)	Median (Ranges)	712.2 (105.9–2511.0)	518.0 (27.8–2632.0)	0.11
** *Urine measurements* **				
**Parameter**	**Statistical Analysis**	**CKD Patients**	**Non-CKD Patients**	***p*-Value**
*Proteinuria (n%)*				
1+	n (%)	33 (47.1%)	2 (4.0%)	<0.001
2+	n (%)	6 (8.6%)	3 (6.0%)	
3+	n (%)	5 (7.1%)	1 (2.0%)	
Neg.	n (%)	26 (37.1%)	44 (88.0%)	
*Hematuria (n%)*				
1+	n (%)	12 (17.1%)	0 (0.0%)	0.005
2+	n (%)	4(5.7%)	1(2.0%)	
3+	n (%)	3(4.3%)	2(4.0%)	
Neg.	n (%)	51(72.9%)	47(94.0%)	
*Leukocyturia (n%)*				
1+	n (%)	5 (7.1%)	0 (0.0%)	0.06
2+	n (%)	0 (0.0%)	2 (4.0%)	
3+	n (%)	2 (2.9%)	1 (2.0%)	
Neg.	n (%)	63 (90.0%)	47 (94.0%)	

Abbreviations: CKD, chronic kidney disease; eGFR, estimated glomerular filtration rate; CRP, C-reactive protein; Neg., negative; SD, standard deviation. ***** denotes that one non-CKD patient presented at hospital admission with acute kidney injury.

**Table 3 jpm-12-01676-t003:** Within-group comparisons and changes in the laboratory findings between admission and day 7–10 follow-up.

Parameter		Groups CKD (*n* = 70) Non-CKD (*n* = 50)	Admission	Control	Change	*p*-Value
**Creatinine (μmol/L)**	Median (ranges)	CKD	119 (57.0–930.0)	100.0 (56.0–716.0)	−17.5 (−331.0–451.0)	0.006
		Non-CKD	79.0 (50–1295.0)	65.0 (46.0–900.0)	−13.5 (−395.0–81.0)	<0.001
**eGFR (mL/min/1.73 m^2^)**	Mean (SD)	CKD	49.5(24.07)	59.0(27.35)	9.5(20.88)	0.01
		Non-CKD	82.3(27.77)	92.2(27.12)	10.0(16.67)	<0.001
**Urea (mmol/L)**	Median (ranges)	CKD	9.2 (2.7–75.2)	8.9 (2.3–60.9)	0.0 (−41.0–44.0)	0.16
		Non-CKD	5.5 (3.0–82.3)	6.0 (3.5–56.9)	0.0 (−25.4–3.4)	0.49
**D-Dimer (mg/L FEU)**	Median (ranges)	CKD	0.9 (0.3–10.7)	0.8 (0.3–10.7)	−0.2 (−4.4–6.1)	0.006
		Non-CKD	0.5 (0.3–8.1)	0.4 (0.3–6.1)	−0.1 (−3.4–0.7)	0.11
**Fibrinogen (g/L)**	Median (ranges)	CKD	4.1 (0.5–12)	4.1 (0.5–7.8)	0.0 (−5.8–3.8)	0.31
		Non-CKD	4.1 (2.0–7.8)	4.0 (2.9–11.1)	0.0 (−3.5–3.8)	0.24
**Hemoglobin (g/L)**	Mean (SD)	CKD	133.7 (19.61)	125.8 (24.17)	−7.9 (17.37)	0.0001
		Non-CKD	142.5 (15.99)	136.1 (18.64)	−6.4 (12.09)	0.01
**Leukocytes (10^3^/mm^3^)**	Median (ranges)	CKD	7.4 (2.5–21.2)	10.4 (1.0–26.0)	1.0 (−8.4–20.6)	0.002
		Non-CKD	6.8 (2.9–35.3)	9.9 (2.9–26.0)	0.7 (−9.3–18.4)	0.03
**Lymphocytes (10^3^/mm^3^)**	Mean (SD)	CKD	1.3 (0.61)	1.3 (0.90)	−0.1 (0.76)	0.03
		Non-CKD	1.4 (0.68)	1.2 (0.73)	0.2 (0.75)	0.04
**Platelets** **(10^3^/mm^3^)**	Mean (SD)	CKD	247.6 (107.31)	271.6 (138.37)	23.9 (112.36)	0.03
		Non-CKD	246.0 (85.14)	291.5 (105.10)	45.5 (137.81)	0.03
**Protein (g/L)**	Mean (SD)	CKD	62.7 (7.09)	59.2 (6.49)	−3.5 (6.09)	<0.001
		Non-CKD	64.1 (6.23)	62.2 (7.56)	−2.0 (5.96)	0.03
**Albumin (g/L)**	Mean (SD)	CKD	36.0 (4.68)	33.5 (4.69)	−2.5 (4.47)	0.0001
		Non-CKD	37.4 (4.91)	36.0 (5.62)	−1.5 (4.81)	0.19
**CRP (mg/L)**	Median (ranges)	CKD	50.8 (0.5–320.6)	17.0 (0.3–242.0)	−26.7 (−227.4–131.0)	0.0006
		Non-CKD	29.8 (0.1–217.6)	4.0 (0.1–186.1)	−21.0 (−215.3–186.0)	0.04
**Ferritin (μg/L)**	Median (ranges)	CKD	712.8 (105.9–2511.0)	852.6 (189.2–3028.3)	96.2 (−989.9–2068.7)	0.0007
		Non-CKD	518.0 (27.8–2632.0)	811.9 (27.8−2535.0)	57.0 (−1085.0–1428.7)	0.28
**Oxygen saturation** **(%)**	Mean (SD)	CKD	92.9 (5.43)	93.8 (3.01)	0.9 (4.67)	0.0002
		Non-CKD	93.8 (6.02)	95.0 (3.18)	1.2 (5.42)	0.003

Abbreviations: CKD, chronic kidney disease; eGFR, estimated glomerular filtration rate; CRP, C-reactive protein; SD, standard deviation.

**Table 4 jpm-12-01676-t004:** Logistic regression model for risk factors for mortality.

Parameter	Class Value for OR Estimate	OR Estimate	Standard Error	Wald Chi-Square	*p*-Value	Exponentiation of OR Estimate
Intercept		−2.3598	0.5342	19.5132	<0.0001	0.094
CKD	CKD	0.6602	0.5406	1.4914	0.22	1.935
AKI	Yes	0.8303	0.3877	4.5873	0.03	2.294
Gender	Female	−0.6721	0.3456	3.7815	0.05	0.511
eGFR	≤45 mL/min/1.73 m^2^	−0.1784	0.4032	0.1958	0.65	0.837
Age	65 y and over	−0.0552	0.3387	0.0266	0.87	0.946
CVD	Yes	−0.1303	0.3369	0.1497	0.69	0.878
Diabetes	Yes	−0.3796	0.3577	1.1257	0.28	0.684
Hypertension	Yes	0.6872	0.5636	1.4868	0.22	1.988
Febrile	Yes	0.4816	0.3305	2.1225	0.14	1.619
Days before hospital	<6 days	0.1877	0.3171	0.3503	0.55	1.206
Remdesivir	Yes	0.8789	0.3402	6.6759	0.009	2.408
Proteinuria	Yes	−0.5885	0.3647	2.6047	0.10	0.555

Abbreviations**:** CKD, chronic kidney disease; AKI, acute kidney infection; CVD, cardiovascular disease; GFR, estimated glomerular filtration; y, years.

**Table 5 jpm-12-01676-t005:** Risk factors for mortality: Odds ratio.

Effect	Odds Ratio	Lower CL *	Upper CL *	*p*-Value
CKD: Yes vs. No	3.745	0.450	31.179	0.22
AKI: Yes vs. No	5.263	1.151	24.056	0.03
Gender: Female vs. Male	0.261	0.067	1.011	0.05
eGFR ≤ 45 mL/min/1.73 m^2^ vs. >45 mL/min/1.73 m^2^	0.700	0.144	3.399	0.65
Age: 65 y and over vs. <65 y	0.895	0.237	3.379	0.87
CVD: Yes vs. No	0.771	0.206	2.886	0.69
Diabetes: Yes vs. No	0.468	0.115	1.903	0.28
Hypertension: Yes vs. No	3.952	0.434	35.995	0.22
Febrile: Yes vs. No	2.620	0.717	9.572	0.14
Days before hospitalization: <6 vs. ≥6	1.456	0.420	5.045	0.55
Remdesivir: Yes vs. No	5.800	1.529	22.005	0.009
Proteinuria: Yes vs. No	0.308	0.074	1.287	0.10

* Wald 95%. Abbreviations: CL, confidence limit for adjusted odds ratio; CKD, chronic kidney disease; AKI, acute kidney infection; GFR, glomerular filtration rate; CVD, cardiovascular disease; eGFR, estimated glomerular filtration rate; y, years.

**Table 6 jpm-12-01676-t006:** Logistic regression model output for acute kidney injury (AKI).

Parameter	Class Value for OR Estimate	DF	OR Estimate	Standard Error	Wald Chi-Square	*p*-Value
Intercept		1	−0.7944	0.3821	4.3217	0.03
CKD	CKD	1	−0.0874	0.5024	0.0302	0.86
eGFR	≤45 mL/min/1.73 m^2^	1	1.3263	0.3251	16.6406	<0.0001
Gender	Female	1	−0.1210	0.2782	0.1892	0.66
Age	65 y and over	1	−0.2291	0.2940	0.6071	0.43
CVD	Yes	1	−0.2859	0.2934	0.9493	0.32
Diabetes	Yes	1	−0.2551	0.3187	0.6408	0.42
Hypertension	Yes	1	0.4261	0.4946	0.7422	0.38
Febrile	Yes	1	0.5497	0.3080	3.1858	0.07
Days before hospital	<6 days	1	0.5449	0.2824	3.7223	0.05
Remdesivir	Yes	1	0.6590	0.2758	5.7092	0.01
Proteinuria	Yes	1	0.4496	0.2952	2.3206	0.12

Abbreviations: CKD, chronic kidney disease; AKI, acute kidney infection; eGFR, estimated glomerular filtration rate; CVD, cardiovascular disease; y, years.

**Table 7 jpm-12-01676-t007:** Logistic regression model: Odds ratios for acute kidney injury (AKI).

Effect	Odds Ratio	Lower CL *	Upper CL *	*p*-Value
CKD: Yes vs. No	0.840	0.117	6.018	0.86
eGFR ≤ 45 mL/min/1.73 m^2^ vs >45 mL/min/1.73 m^2^	14.191	3.967	50.762	<0.0001
Gender: Female vs. Male	0.785	0.264	2.336	0.66
Age: 65 y and over vs. <65 y	0.632	0.200	2.002	0.43
CVD: Yes vs. No	0.565	0.179	1.783	0.32
Diabetes: Yes vs. No	0.600	0.172	2.094	0.42
Hypertension: Yes vs. No	2.345	0.337	16.297	0.38
Febrile: Yes vs. No	3.003	0.898	10.042	0.07
Days before hospitalization: <6 vs. ≥6	2.974	0.983	8.998	0.05
Remdesivir: Yes vs. No	3.736	1.267	11.015	0.01
Proteinuria: Yes vs. No	2.458	0.773	7.817	0.12

* Wald 95%. Abbreviations: CL, confidence limit for adjusted odds ratio; CKD, chronic kidney disease; AKI, acute kidney infection; CVD, cardiovascular disease; eGFR, estimated glomerular filtration; y, years.

## Data Availability

Data are available upon request.

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
