# Peer review of "COVID-19 Infection in Chronic Kidney Disease Patients in Bulgaria: Risk Factors for Death and Acute Kidney Injury"

_jpm, 2022, doi:10.3390/jpm12101676_

Round 1

Reviewer 1 Report

- In the CKD group (impaired kidney function with estimated glomerular filtration rate of <60 ml/min), the initial eGFR was 49.5(24.07) ml/min and 59.0(27.35) after 7-10 days. There are therefore some patients who have an eGFR greater than 60 ml/min at 7-10 days. These patients do not meet the definition criterion of CKD group (eGFR<60). 

-The control group included five patients that had undergone renal transplantation. This is controversial even with an eGFR greater than 60 ml/min. The authors should provide more details on these patients (post-transplant time...) and their specific fate.

- The use of the average eGFR is open to criticism. Indeed, in each of the two groups, there are patients in AKI to whom the eGFR cannot be applied.

In acute setting what is the validity of using eGFR calculation?

- All non-Caucasian patients (n=21) are in the CKD group. This point must be addressed in the discussion. 

- In Tables 1 and 2, we recommend using the same order of CKD and not CKD  columns. This will avoid any confusion. 

- The expression of certain parameters (ferritinemia, CRP ...) in average is not enough. It would be interesting to use the percentage of patients outside the normal range. 

Author Response

We want to thank Reviewer #1 for his/her time to review our manuscript and we thank him/her for the comments – positive and constructive questions/comments.

- In the CKD group (impaired kidney function with estimated glomerular filtration rate of <60 ml/min), the initial eGFR was 49.5(24.07) ml/min and 59.0(27.35) after 7-10 days. There are therefore some patients who have an eGFR greater than 60 ml/min at 7-10 days. These patients do not meet the definition criterion of CKD group (eGFR<60).

ANSWER:
Indeed, the wording was not very clear. CKD was based on the KDIGO Stages that were defined by the guidelines. Therefore, CKD group included patients with eGFR higher than 60 ml/min/1.73m2. The five kidney transplant patients were misclassified and indeed they belong to the CKD group. In the manuscript we made the modifications accordingly (see Page 2, Section “Patients and methods”).

-The control group included five patients that had undergone renal transplantation. This is controversial even with an eGFR greater than 60 ml/min. The authors should provide more details on these patients (post-transplant time...) and their specific fate.

ANSWER:

Indeed, the five kidney transplant patients were misclassified (see above): we reclassified them. All the patients were transplanted for more than 5 years (see Page 2, Section “Patients and methods”). All the transplanted patients had died and this is now mentioned (see Page 7, Section “Results”, line 225).

- The use of the average eGFR is open to criticism. Indeed, in each of the two groups, there are patients in AKI to whom the eGFR cannot be applied.

ANSWER:

Indeed, we agree on that point but it is the only way to address the renal function: we therefore applied the KDIGO guidelines for patients with AKI. We fully understand the criticisms and the limitations of estimating the eGFR in the settings of AKI. However,  we already specified in the previous version of our manuscript that the incidence of AKI was assessed by using the KDIGO criteria by calculating kidney function from creatinine levels (Page 3, Section “Patients and Methods”, line 110)

In acute setting what is the validity of using eGFR calculation?

ANSWER:

This is the limitation of the eGFR in this setting, but we cannot do better. See above.

- All non-Caucasian patients (n=21) are in the CKD group. This point must be addressed in the discussion.

In the reviewed paper we have not stated that there are any non-Caucasian patients – in Table 1 we have showed that from the two groups 100% of the patients are Caucasians. To be clearer we have added a sentence stating that all patients are Caucasians. (Page 2, Section “Patients and Methods”, line 81).

- In Tables 1 and 2, we recommend using the same order of CKD and not CKD columns. This will avoid any confusion.

ANSWER:

Thank you for the suggestion that we took into account. Therefore Table 1 has been modified accordingly (See Page 4, Section “Results”, Line 171).

- The expression of certain parameters (ferritinemia, CRP ...) in average is not enough. It would be interesting to use the percentage of patients outside the normal range.

ANSWER:

Thank you for the suggestion that we took into account. On admission 23 patients (19.1%) had normal levels of CRP, 66 patients (55%) had normal range of serum ferritin, 36 patients (30%) had normal levels of D-dimers and finally for fibrinogen 60 patients (50%) had normal levels. (Page 4, Section “Results”, Line 177).

Reviewer 2 Report

Well presented and written. Should be considered for publication. 

Author Response

Comment: Well presented and written. Should be considered for publication. 

We want to thank Reviewer #2 for his/her time to review our manuscript and we thank him/her for the positive comments and the consideration for the publication.

Reviewer 3 Report

This is a good articule that contributes ti the knowledge of te disease 

Author Response

Comment: This is a good article that contributes the knowledge of the disease.

We want to thank Reviewer #3 for his/her time to review our manuscript and we thank him/her for the positive comments and acknowledge that it will contribute to the understanding of the disease. Thank you for the consideration for the publication.